# Phase Retrieval using Alternating Minimization

**Praneeth Netrapalli**
Department of ECE
The University of Texas at Austin
Austin, TX 78712
praneethn@utexas.edu

**Prateek Jain**
Microsoft Research India
Bangalore, India
prajain@microsoft.com

**Sujay Sanghavi**
Department of ECE
The University of Texas at Austin
Austin, TX 78712
sanghavi@mail.utexas.edu

## Abstract

Phase retrieval problems involve solving linear equations, but with missing sign (or phase, for complex numbers). Over the last two decades, a popular generic empirical approach to the many variants of this problem has been one of alternating minimization; i.e. alternating between estimating the missing phase information, and the candidate solution. In this paper, we show that a simple alternating minimization algorithm geometrically converges to the solution of one such problem – finding a vector $\mathbf{x}$ from $\mathbf{y}, \mathbf{A}$, where $\mathbf{y} = |\mathbf{A}^T\mathbf{x}|$ and $|\mathbf{z}|$ denotes a vector of element-wise magnitudes of $\mathbf{z}$ – under the assumption that $\mathbf{A}$ is Gaussian.

Empirically, our algorithm performs similar to recently proposed convex techniques for this variant (which are based on "lifting" to a convex matrix problem) in sample complexity and robustness to noise. However, our algorithm is much more efficient and can scale to large problems. Analytically, we show geometric convergence to the solution, and sample complexity that is off by log factors from obvious lower bounds. We also establish close to optimal scaling for the case when the unknown vector is sparse. Our work represents the only known theoretical guarantee for alternating minimization for any variant of phase retrieval problems in the non-convex setting.

## 1  Introduction

In this paper we are interested in recovering a complex[1] vector $\mathbf{x}^* \in \mathbb{C}^n$ from *magnitudes of* its linear measurements. That is, for $\mathbf{a_i} \in \mathbb{C}^n$, if

$$y_i = |\langle \mathbf{a_i}, \mathbf{x}^* \rangle|, \quad \text{for } i = 1, \dots, m \tag{1}$$

then the task is to recover $\mathbf{x}^*$ using $\mathbf{y}$ and the measurement matrix $\mathbf{A} = [\mathbf{a_1} \ \mathbf{a_2} \ \dots \ \mathbf{a_m}]$.

The above problem arises in many settings where it is harder / infeasible to record the phase of measurements, while recording the magnitudes is significantly easier. This problem, known as *phase retrieval*, is encountered in several applications in crystallography, optics, spectroscopy and tomography [14]. Moreover, the problem is broadly studied in the following two settings:

   (i) The measurements in (1) correspond to the Fourier transform (the number of measurements here is equal to $n$) and there is some apriori information about the signal.

(ii) The set of measurements $\mathbf{y}$ are overcomplete (i.e., $m > n$), while some apriori information about the signal may or may not be available.

In the first case, various types of apriori information about the underlying signal such as positivity, magnitude information on the signal [11], sparsity [25] and so on have been studied. In the second case, algorithms for various measurement schemes such as Fourier oversampling [21], multiple random illuminations [4, 28] and wavelet transform [28] have been suggested.

By and large, the most well known methods for solving this problem are the error reduction algorithms due to Gerchberg and Saxton [13] and Fienup [11], and variants thereof. These algorithms are alternating projection algorithms that iterate between the unknown phases of the measurements and the unknown underlying vector. Though the empirical performance of these algorithms has been well studied [11, 19], and they are used in many applications [20], there are not many theoretical guarantees regarding their performance.

More recently, a line of work [7, 6, 28] has approached this problem from a different angle, based on the realization that recovering $\mathbf{x}^*$ is equivalent to recovering the rank-one matrix $\mathbf{x}^*\mathbf{x}^{*T}$, i.e., its outer product. Inspired by the recent literature on trace norm relaxation of the rank constraint, they design SDPs to solve this problem. Refer Section 1.1 for more details.

**In this paper** we go back to the empirically more popular ideology of alternating minimization; we develop a new alternating minimization algorithm, for which we show that *(a)* empirically, it noticeably outperforms convex methods, and *(b)* analytically, a natural resampled version of this algorithm requires $O(n \log^3 n)$ i.i.d. random Gaussian measurements to geometrically converge to the true vector.

**Our contribution**:

- The iterative part of our algorithm is implicit in previous work [13, 11, 28, 4]; the novelty in our *algorithmic contribution* is the initialization step which makes it more likely for the iterative procedure to succeed - see Figures 1 and 2.

- Our *analytical contribution* is the *first theoretical guarantee* regarding the convergence of alternating minimization for the phase retrieval problem in a non-convex setting.

- When the underlying vector is *sparse*, we design another algorithm that achieves a sample complexity of $O\left((x^*_{\min})^{-4} \left(\log n + \log^3 k\right)\right)$ where $k$ is the sparsity and $x^*_{\min}$ is the minimum non-zero entry of $\mathbf{x}^*$. This algorithm also runs over $\mathbb{C}^n$ and scales much better than SDP based methods.

Besides being an empirically better algorithm for this problem, our work is also interesting in a broader sense: there are many problems in machine learning where the natural formulation of a problem is non-convex; examples include rank constrained problems, applications of EM algorithms etc., and alternating minimization has good empirical performance. However, the methods with the best (or only) analytical guarantees involve convex relaxations (e.g., by relaxing the rank constraint and penalizing the trace norm). In most of these settings, correctness of alternating minimization is an open question. We believe that our results in this paper are of interest, and may have implications, in this larger context.

The rest of the paper is organized as follows: In section 1.1, we briefly review related work. We clarify our notation in Section 2. We present our algorithm in Section 3 and the main results in Section 4. We present our results for the sparse case in Section 5. Finally, we present experimental results in Section 6.

## 1.1 Related Work

**Phase Retrieval via Non-Convex Procedures**: Inspite of the huge amount of work it has attracted, phase retrieval has been a long standing open problem. Early work in this area focused on using holography to capture the phase information along with magnitude measurements [12]. However, computational methods for reconstruction of the signal using only magnitude measurements received a lot of attention due to their applicability in resolving spurious noise, fringes, optical system aberrations and so on and difficulties in the implementation of interferometer setups [9]. Though such methods have been developed to solve this problem in various practical settings [8, 20], our

theoretical understanding of this problem is still far from complete. Many papers have focused on determining conditions under which (1) has a unique solution - see [24] and references therein. However, the uniqueness results of these papers do not resolve the algorithmic question of how to find the solution to (1).

Since the seminal work of Gerchberg and Saxton [13] and Fienup [11], many iterated projection algorithms have been developed targeted towards various applications [1, 10, 2]. [21] first suggested the use of multiple magnitude measurements to resolve the phase problem. This approach has been successfully used in many practical applications - see [9] and references there in. Following the empirical success of these algorithms, researchers were able to explain its success in some of the instances [29] using Bregman's theory of iterated projections onto convex sets [3]. However, many instances, such as the one we consider in this paper, are out of reach of this theory since they involve magnitude constraints which are non-convex. To the best of our knowledge, there are no theoretical results on the convergence of these approaches in a non-convex setting.

**Phase Retrieval via Convex Relaxation**: An interesting recent approach for solving this problem formulates it as one of finding the rank-one solution to a system of linear matrix equations. The papers [7, 6] then take the approach of relaxing the rank constraint by a trace norm penalty, making the overall algorithm a convex program (called *PhaseLift*) over $n \times n$ matrices. Another recent line of work [28] takes a similar but different approach : it uses an SDP relaxation (called *PhaseCut*) that is inspired by the classical SDP relaxation for the max-cut problem. To date, these convex methods are the only ones with analytical guarantees on statistical performance [5, 28] (i.e. the number $m$ of measurements required to recover $\mathbf{x}^*$) – under an i.i.d. random Gaussian model on the measurement vectors $\mathbf{a}_i$. However, by "lifting" a vector problem to a matrix one, these methods lead to a much larger representation of the state space, and higher computational cost as a result.

**Sparse Phase Retrieval**: A special case of the phase retrieval problem which has received a lot of attention recently is when the underlying signal $\mathbf{x}^*$ is known to be sparse. Though this problem is closely related to the compressed sensing problem, lack of phase information makes this harder. However, the $\ell_1$ regularization approach of compressed sensing has been successfully used in this setting as well. In particular, if $\mathbf{x}^*$ is sparse, then the corresponding lifted matrix $\mathbf{x}^*\mathbf{x}^{*T}$ is also sparse. [22, 18] use this observation to design $\ell_1$ regularized SDP algorithms for phase retrieval of sparse vectors. For random Gaussian measurements, [18] shows that $\ell_1$ regularized PhaseLift recovers $\mathbf{x}^*$ correctly if the number of measurements is $\Omega(k^2 \log n)$. By the results of [23], this result is tight up to logarithmic factors for $\ell_1$ and trace norm regularized SDP relaxations.

**Alternating Minimization** (a.k.a. **ALS**): Alternating minimization has been successfully applied to many applications in the low-rank matrix setting. For example, clustering, sparse PCA, non-negative matrix factorization, signed network prediction etc. - see [15] and references there in. However, despite empirical success, for most of the problems, there are no theoretical guarantees regarding its convergence except to a local minimum. The only exceptions are the results in [16, 15] which give provable guarantees for alternating minimization for the problems of matrix sensing and matrix completion.

## 2 Notation

We use bold capital letters ($\mathbf{A}, \mathbf{B}$ etc.) for matrices, bold small case letters ($\mathbf{x}, \mathbf{y}$ etc.) for vectors and non-bold letters ($\alpha, U$ etc.) for scalars. For every complex vector $\mathbf{w} \in \mathbb{C}^n$, $|\mathbf{w}| \in \mathbb{R}^n$ denotes its element-wise magnitude vector. $\mathbf{w}^T$ and $\mathbf{A}^T$ denote the Hermitian transpose of the vector $\mathbf{w}$ and the matrix $\mathbf{A}$ respectively. $\mathbf{e_1}, \mathbf{e_2}$, etc. denote the canonical basis vectors in $\mathbb{C}^n$. $\overline{z}$ denotes the complex conjugate of the complex number $z$. In this paper we use the standard Gaussian (or normal) distribution over $\mathbb{C}^n$. $\mathbf{a}$ is said to be distributed according to this distribution if $\mathbf{a} = \mathbf{a_1} + i\mathbf{a_2}$, where $\mathbf{a_1}$ and $\mathbf{a_2}$ are independent and are distributed according to $\mathcal{N}(0, \mathbf{I})$. We also define $\mathrm{Ph}(z) \overset{\text{def}}{=} \frac{z}{|z|}$ for every $z \in \mathbb{C}$, and $\mathrm{dist}(\mathbf{w_1}, \mathbf{w_2}) \overset{\text{def}}{=} \sqrt{1 - \left|\frac{\langle \mathbf{w_1}, \mathbf{w_2} \rangle}{\|\mathbf{w_1}\|_2 \|\mathbf{w_2}\|_2}\right|^2}$ for every $\mathbf{w_1}, \mathbf{w_2} \in \mathbb{C}^n$. Finally, we use the shorthand wlog for without loss of generality and whp for with high probability.

## 3 Algorithm

In this section, we present our alternating minimization based algorithm for solving the phase retrieval problem. Let $\mathbf{A} \in \mathbb{C}^{n \times m}$ be the measurement matrix, with $\mathbf{a}_i$ as its $i^{th}$ column; similarly let

---

**Algorithm 1** AltMinPhase

---

**input** $\mathbf{A}, \mathbf{y}, t_0$
  1: Initialize $\mathbf{x^0} \leftarrow$ top singular vector of $\sum_i y_i^2 \mathbf{a_i a_i}^T$
  2: **for** $t = 0, \cdots, t_0 - 1$ **do**
  3:    $\mathbf{C^{t+1}} \leftarrow \mathrm{Diag}\left(\mathrm{Ph}\left(\mathbf{A}^T \mathbf{x^t}\right)\right)$
  4:    $\mathbf{x^{t+1}} \leftarrow \mathrm{argmin}_{\mathbf{x} \in \mathbb{R}^n} \left\|\mathbf{A}^T \mathbf{x} - \mathbf{C^{t+1}} \mathbf{y}\right\|_2$
  5: **end for**
**output** $\mathbf{x^{t_0}}$

---

$\mathbf{y}$ be the vector of recorded magnitudes. Then,

$$\mathbf{y} = |\mathbf{A}^T \mathbf{x}^*|.$$

Recall that, given $\mathbf{y}$ and $\mathbf{A}$, the goal is to recover $\mathbf{x}^*$. If we had access to the true phase $\mathbf{c}^*$ of $A^T \mathbf{x}^*$ (i.e., $c_i^* = \mathrm{Ph}\left(\langle \mathbf{a_i}, \mathbf{x}^* \rangle\right)$) and $m \geq n$, then our problem reduces to one of solving a system of linear equations:

$$\mathbf{C}^* \mathbf{y} = \mathbf{A}^T \mathbf{x}^*,$$

where $\mathbf{C}^* \overset{\mathrm{def}}{=} \mathrm{Diag}(\mathbf{c}^*)$ is the diagonal matrix of phases. Of course we do not know $\mathbf{C}^*$, hence one approach to recovering $\mathbf{x}^*$ is to solve:

$$\underset{\mathbf{C},\mathbf{x}}{\mathrm{argmin}} \ \|\mathbf{A}^T \mathbf{x} - \mathbf{C} \mathbf{y}\|_2, \tag{2}$$

where $\mathbf{x} \in \mathbb{C}^n$ and $\mathbf{C} \in \mathbb{C}^{m \times m}$ is a diagonal matrix with each diagonal entry of magnitude 1. Note that the above problem is *not convex* since $\mathbf{C}$ is restricted to be a diagonal phase matrix and hence, one cannot use standard convex optimization methods to solve it.

Instead, our algorithm uses the well-known alternating minimization: alternatingly update $\mathbf{x}$ and $\mathbf{C}$ so as to minimize (2). Note that given $\mathbf{C}$, the vector $\mathbf{x}$ can be obtained by solving the following least squares problem: $\min_{\mathbf{x}} \|\mathbf{A}^T \mathbf{x} - \mathbf{C} \mathbf{y}\|_2$. Since the number of measurements $m$ is larger than the dimensionality $n$ and since each entry of $\mathbf{A}$ is sampled from independent Gaussians, $\mathbf{A}$ is invertible with probability 1. Hence, the above least squares problem has a unique solution. On the other hand, given $\mathbf{x}$, the optimal $\mathbf{C}$ is given by $\mathbf{C} = \mathrm{Diag}\left(\mathrm{Ph}\left(\mathbf{A}^T \mathbf{x}\right)\right)$.

While the above algorithm is simple and intuitive, it is known that with bad initial points, the solution might not converge to $\mathbf{x}^*$. In fact, this algorithm with a uniformly random initial point has been empirically evaluated for example in [28], where it performs worse than SDP based methods. Moreover, since the underlying problem is non-convex, standard analysis techniques fail to guarantee convergence to the global optimum, $\mathbf{x}^*$. Hence, the key challenges here are: a) a good initialization step for this method, b) establishing this method's convergence to $\mathbf{x}^*$.

We address the first key challenge in our AltMinPhase algorithm (Algorithm 1) by initializing $\mathbf{x}$ as the largest singular vector of the matrix $\mathbf{S} = \frac{1}{m} \sum_i y_i^2 \mathbf{a_i a_i}^T$. Theorem 4.1 shows that when $\mathbf{A}$ is sampled from standard complex normal distribution, this initialization is accurate. In particular, if $m \geq C_1 n \log^3 n$ for large enough $C_1 > 0$, then whp we have $\|\mathbf{x^0} - \mathbf{x}^*\|_2 \leq 1/100$ (or any other constant).

Theorem 4.2 addresses the second key challenge and shows that a variant of AltMinPhase (see Algorithm 2) actually converges to the global optimum $\mathbf{x}^*$ at linear rate. See section 4 for a detailed analysis of our algorithm.

We would like to stress that not only does a natural variant of our proposed algorithm have rigorous theoretical guarantees, it also is effective practically as each of its iterations is fast, has a closed form solution and does not require SVD computation. AltMinPhase has similar statistical complexity to that of PhaseLift and PhaseCut while being much more efficient computationally. In particular, for accuracy $\epsilon$, we only need to solve each least squares problem only up to accuracy $O(\epsilon)$. Now, since the measurement matrix $A$ is sampled from Gaussian with $m > Cn$, it is well conditioned. Hence, using conjugate gradient method, each such step takes $O\left(mn \log \frac{1}{\epsilon}\right)$ time. When $m = O(n)$ and we have geometric convergence, the total time taken by the algorithm is $O\left(n^2 \log^2 \frac{1}{\epsilon}\right)$. SDP based methods on the other hand require $\Omega(n^3/\sqrt{\epsilon})$ time. Moreover, our initialization step increases the likelihood of successful recovery as opposed to a random initialization (which has been considered so far in prior work). Refer Figure 1 for an empirical validation of these claims.

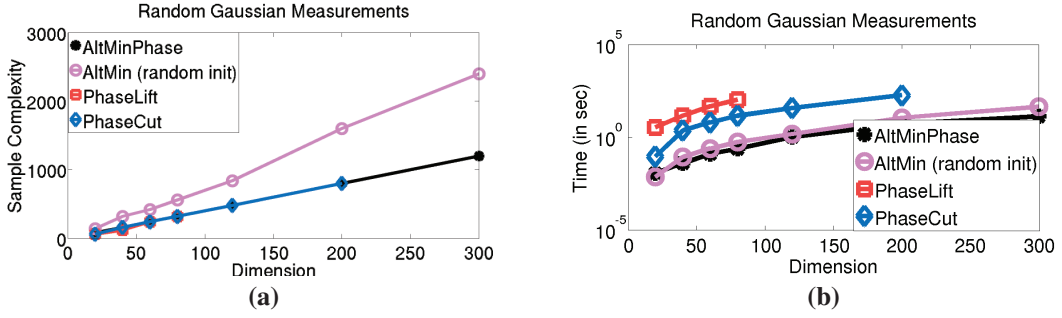

Figure 1: Sample and Time complexity of various methods for Gaussian measurement matrices $A$. Figure 1(a) compares the number of measurements required for successful recovery by various methods. We note that our initialization improves sample complexity over that of random initialization (AltMin (random init)) by a factor of 2. AltMinPhase requires similar number of measurements as PhaseLift and PhaseCut. Figure 1(b) compares the running time of various algorithms on log-scale. Note that AltMinPhase is almost two orders of magnitude faster than PhaseLift and PhaseCut.

## 4 Main Results: Analysis

In this section we describe the main contribution of this paper: provable statistical guarantees for the success of alternating minimization in solving the phase recovery problem. To this end, we consider the setting where each measurement vector $\mathbf{a_i}$ is iid and is sampled from the standard complex normal distribution. We would like to stress that all the existing guarantees for phase recovery also use exactly the same setting [6, 5, 28]. Table 1 presents a comparison of the theoretical guarantees of Algorithm 2 as compared to PhaseLift and PhaseCut.

| | Sample complexity | Comp. complexity |
|---|---|---|
| Algorithm 2 | $O\left(n\left(\log^3 n + \log\frac{1}{\epsilon}\log\log\frac{1}{\epsilon}\right)\right)$ | $O\left(n^2\left(\log^3 n + \log^2\frac{1}{\epsilon}\log\log\frac{1}{\epsilon}\right)\right)$ |
| PhaseLift [5] | $O\left(n\right)$ | $O\left(n^3/\epsilon^2\right)$ |
| PhaseCut [28] | $O\left(n\right)$ | $O\left(n^3/\sqrt{\epsilon}\right)$ |

Table 1: Comparison of Algorithm 2 with PhaseLift and PhaseCut: Though the sample complexity of Algorithm 2 is off by $\log$ factors from that of PhaseLift and PhaseCut, it is $O\left(n\right)$ better than them in computational complexity. Note that, we can solve the least squares problem in each iteration approximately by using conjugate gradient method which requires only $O\left(mn\right)$ time.

Our proof for convergence of alternating minimization can be broken into two key results. We first show that if $m \geq Cn\log^3 n$, then whp the initialization step used by AltMinPhase returns $\mathbf{x^0}$ which is at most a constant distance away from $\mathbf{x}^*$. Furthermore, that constant can be controlled by using more samples (see Theorem 4.1).

We then show that if $\mathbf{x^t}$ is a *fixed* vector such that $\text{dist}\left(\mathbf{x^t}, \mathbf{x}^*\right) < c$ (small enough) and $\mathbf{A}$ is sampled independently of $\mathbf{x^t}$ with $m > Cn$ ($C$ large enough) then whp $\mathbf{x^{t+1}}$ satisfies: $\text{dist}\left(\mathbf{x^{t+1}}, \mathbf{x}^*\right) < \frac{3}{4}\text{dist}\left(\mathbf{x^t}, \mathbf{x}^*\right)$ (see Theorem 4.2). Note that our analysis critically requires $\mathbf{x^t}$ to be "fixed" and be independent of the sample matrix $\mathbf{A}$. Hence, we cannot re-use the same $\mathbf{A}$ in each iteration; instead, we need to resample $\mathbf{A}$ in every iteration. Using these results, we prove the correctness of Algorithm 2, which is a natural resampled version of AltMinPhase.

We now present the two results mentioned above. For our proofs, wlog, we assume that $\|\mathbf{x}^*\|_2 = 1$.

Our first result guarantees a good initial vector.

**Theorem 4.1.** *There exists a constant $C_1$ such that if $m > \frac{C_1}{c^2}n\log^3 n$, then in Algorithm 2, with probability greater than $1 - 4/m^2$ we have:*

$$\|\mathbf{x^0} - \mathbf{x}^*\|_2 < c.$$

---

**Algorithm 2** AltMinPhase with Resampling

---

**input** $\mathbf{A}, \mathbf{y}, \epsilon$

1: $t_0 \leftarrow c \log \frac{1}{\epsilon}$

2: Partition $\mathbf{y}$ and (the corresponding columns of) $A$ into $t_0 + 1$ equal disjoint sets: $(\mathbf{y^0}, \mathbf{A^0}), (\mathbf{y^1}, \mathbf{A^1}), \cdots, (\mathbf{y^{to}}, \mathbf{A^{to}})$

3: $\mathbf{x^0} \leftarrow$ top singular vector of $\sum_l \left(y_l^0\right)^2 \mathbf{a_\ell^0} \left(\mathbf{a_\ell^0}\right)^T$

4: **for** $t = 0, \cdots, t_0 - 1$ **do**

5: $\quad \mathbf{C^{t+1}} \leftarrow \text{Diag}\left(\text{Ph}\left(\left(\mathbf{A^{t+1}}\right)^T \mathbf{x^t}\right)\right)$

6: $\quad \mathbf{x^{t+1}} \leftarrow \text{argmin}_{\mathbf{x} \in \mathbb{R}^n} \left\|\left(\mathbf{A^{t+1}}\right)^T \mathbf{x} - \mathbf{C^{t+1}} \mathbf{y^{t+1}}\right\|_2$

7: **end for**

**output** $\mathbf{x^{to}}$

---

The second result proves geometric decay of error assuming a good initialization.

**Theorem 4.2.** *There exist constants* $c$, $\widehat{c}$ *and* $\widetilde{c}$ *such that in iteration* $t$ *of Algorithm 2, if* $\text{dist}\left(\mathbf{x^t}, \mathbf{x^*}\right) < c$ *and the number of columns of* $\mathbf{A^t}$ *is greater than* $\widehat{c} n \log \frac{1}{\eta}$ *then, with probability more than* $1 - \eta$, *we have:*

$$\text{dist}\left(\mathbf{x^{t+1}}, \mathbf{x^*}\right) < \frac{3}{4} \text{dist}\left(\mathbf{x^t}, \mathbf{x^*}\right), \text{ and } \|\mathbf{x^{t+1}} - \mathbf{x^*}\|_2 < \widetilde{c} \,\text{dist}\left(\mathbf{x^t}, \mathbf{x^*}\right).$$

*Proof.* For simplicity of notation in the proof of the theorem, we will use $\mathbf{A}$ for $\mathbf{A^{t+1}}$, $\mathbf{C}$ for $\mathbf{C^{t+1}}$, $\mathbf{x}$ for $\mathbf{x^t}$, $\mathbf{x^+}$ for $\mathbf{x^{t+1}}$, and $\mathbf{y}$ for $\mathbf{y^{t+1}}$. Now consider the update in the $(t+1)^{\text{th}}$ iteration:

$$\mathbf{x^+} = \underset{\widetilde{\mathbf{x}} \in \mathbb{R}^n}{\text{argmin}} \left\|\mathbf{A}^T \widetilde{\mathbf{x}} - \mathbf{Cy}\right\|_2 = \left(\mathbf{AA}^T\right)^{-1} \mathbf{ACy} = \left(\mathbf{AA}^T\right)^{-1} \mathbf{ADA}^T \mathbf{x^*}, \qquad (3)$$

where $\mathbf{D}$ is a diagonal matrix with $D_{ll} \overset{\text{def}}{=} \text{Ph}\left(\mathbf{a}_\ell{}^T \mathbf{x} \cdot \overline{\mathbf{a}_\ell{}^T \mathbf{x^*}}\right)$. Now (3) can be rewritten as:

$$\mathbf{x^+} = \left(\mathbf{AA}^T\right)^{-1} \mathbf{ADA}^T \mathbf{x^*} = \mathbf{x^*} + \left(\mathbf{AA}^T\right)^{-1} \mathbf{A}\left(\mathbf{D} - \mathbf{I}\right)\mathbf{A}^T \mathbf{x^*}, \qquad (4)$$

that is, $\mathbf{x^+}$ can be viewed as a perturbation of $\mathbf{x^*}$ and the goal is to bound the error term (the second term above). We break the proof into two main steps:

1. $\exists$ a constant $c_1$ such that $|\langle \mathbf{x^*}, \mathbf{x^+}\rangle| \geq 1 - c_1 \text{dist}\left(\mathbf{x}, \mathbf{x^*}\right)$ (see Lemma A.2), and

2. $|\langle \mathbf{z}, \mathbf{x^+}\rangle| \leq \frac{5}{9} \text{dist}\left(\mathbf{x}, \mathbf{x^*}\right)$, for all $\mathbf{z}$ s.t. $\mathbf{z}^T \mathbf{x^*} = 0$. (see Lemma A.4)

Assuming the above two bounds and choosing $c < \frac{1}{100 c_1}$, we can prove the theorem:

$$\text{dist}\left(\mathbf{x^+}, \mathbf{x^*}\right)^2 < \frac{(25/81) \cdot \text{dist}\left(\mathbf{x}, \mathbf{x^*}\right)^2}{(1 - c_1 \text{dist}\left(\mathbf{x}, \mathbf{x^*}\right))^2} \leq \frac{9}{16} \text{dist}\left(\mathbf{x}, \mathbf{x^*}\right)^2,$$

proving the first part of the theorem. The second part follows easily from (3) and Lemma A.2. $\quad\square$

**Intuition and key challenge**: If we look at step 6 of Algorithm 2, we see that, for the measurements, we use magnitudes calculated from $\mathbf{x^*}$ and phases calculated from $\mathbf{x}$. Intuitively, this means that we are trying to push $\mathbf{x^+}$ towards $\mathbf{x^*}$ (since we use its magnitudes) and $\mathbf{x}$ (since we use its phases) at the same time. The key intuition behind the success of this procedure is that the push towards $\mathbf{x^*}$ is stronger than the push towards $\mathbf{x}$, when $\mathbf{x}$ is close to $\mathbf{x^*}$. The key lemma that captures this effect is stated below:

**Lemma 4.3.** *Let* $w_1$ *and* $w_2$ *be two independent standard complex Gaussian random variables[2]. Let* $U = |w_1| w_2 \left(\text{Ph}\left(1 + \frac{\sqrt{1-\alpha^2}\overline{w_2}}{\alpha |w_1|}\right) - 1\right)$. *Fix* $\delta > 0$. *Then, there exists a constant* $\gamma > 0$ *such that if* $\sqrt{1-\alpha^2} < \gamma$, *then:* $\mathbb{E}\left[U\right] \leq (1+\delta)\sqrt{1-\alpha^2}$.

**Algorithm 3** SparseAltMinPhase

---

**input** $\mathbf{A}, \mathbf{y}, k$
1: $S \leftarrow$ top-$k$ $\mathrm{argmax}_{j \in [n]} \sum_{i=1}^{m} |a_{ij} y_i|$ {Pick indices of $k$ largest absolute value inner product}
2: Apply Algorithm 2 on $\mathbf{A}_S, \mathbf{y}_S$ and output the resulting vector with elements in $S^c$ set to zero.

---

| | Sample complexity | Comp. complexity |
|---|---|---|
| Algorithm 3 | $O\left(k\left(k\log n + \log\frac{1}{\epsilon}\log\log\frac{1}{\epsilon}\right)\right)$ | $O\left(k^2\left(kn\log n + \log^2\frac{1}{\epsilon}\log\log\frac{1}{\epsilon}\right)\right)$ |
| $\ell_1$-PhaseLift [18] | $O\left(k^2\log n\right)$ | $O\left(n^3/\epsilon^2\right)$ |

Table 2: Comparison of Algorithm 3 with $\ell_1$-PhaseLift when $x_{\min}^* = \Omega\left(1/\sqrt{k}\right)$. Note that the complexity of Algorithm 3 is dominated by the support finding step. If $k = O(1)$, Algorithm 3 runs in quasi-linear time.

See Appendix A for a proof of the above lemma and how we use it to prove Theorem 4.2.

Combining Theorems 4.1 and 4.2, and a simple observation that $\|\mathbf{x^T} - \mathbf{x}^*\|_2 < \widetilde{c}\,\mathrm{dist}\left(\mathbf{x^T}, \mathbf{x}^*\right)$ for a constant $\widetilde{c}$, we can establish the correctness of Algorithm 2.

**Theorem 4.4.** *Suppose the measurement vectors in (1) are independent standard complex normal vectors. For every $\eta > 0$, there exists a constant $c$ such that if $m > cn\left(\log^3 n + \log\frac{1}{\epsilon}\log\log\frac{1}{\epsilon}\right)$ then, with probability greater than $1 - \eta$, Algorithm 2 outputs $\mathbf{x^{to}}$ such that $\|\mathbf{x^{to}} - \mathbf{x}^*\|_2 < \epsilon$.*

## 5 Sparse Phase Retrieval

In this section, we consider the case where $\mathbf{x}^*$ is known to be sparse, with sparsity $k$. A natural and practical question to ask here is: can the sample and computational complexity of the recovery algorithm be improved when $k \ll n$.

Recently, [18] studied this problem for Gaussian $\mathbf{A}$ and showed that for $\ell_1$ regularized PhaseLift, $m = O(k^2 \log n)$ samples suffice for exact recovery of $\mathbf{x}^*$. However, the computational complexity of this algorithm is still $O(n^3/\epsilon^2)$.

In this section, we provide a simple extension of our AltMinPhase algorithm that we call SparseAlt-MinPhase, for the case of sparse $\mathbf{x}^*$. The main idea behind our algorithm is to first recover the support of $\mathbf{x}^*$. Then, the problem reduces to phase retrieval of a $k$-dimensional signal. We then solve the reduced problem using Algorithm 2. The pseudocode for SparseAltMinPhase is presented in Algorithm 3. Table 2 provides a comparison of Algorithm 3 with $\ell_1$-regularized PhaseLift in terms of sample complexity as well as computational complexity.

The following lemma shows that if the number of measurements is large enough, step 1 of SparseAlt-MinPhase recovers the support of $\mathbf{x}^*$ correctly.

**Lemma 5.1.** *Suppose $\mathbf{x}^*$ is $k$-sparse with support $S$ and $\|\mathbf{x}^*\|_2 = 1$. If $\mathbf{a_i}$ are standard complex Gaussian random vectors and $m > \frac{c}{\left(x_{min}^*\right)^4}\log\frac{n}{\delta}$, then Algorithm 3 recovers $S$ with probability greater than $1 - \delta$, where $x_{min}^*$ is the minimum non-zero entry of $\mathbf{x}^*$.*

The key step of our proof is to show that if $j \in supp(\mathbf{x}^*)$, then random variable $Z_{ij} = \sum_i |a_{ij} y_i|$ has significantly higher mean than for the case when $j \notin supp(\mathbf{x}^*)$. Now, by applying appropriate concentration bounds, we can ensure that $\min_{j \in supp(\mathbf{x}^*)} |Z_{ij}| > \max_{j \notin supp(\mathbf{x}^*)} |Z_{ij}|$ and hence our algorithm never picks up an element outside the true support set $supp(\mathbf{x}^*)$. See Appendix B for a detailed proof of the above lemma.

The correctness of Algorithm 3 now is a direct consequence of Lemma 5.1 and Theorem 4.4. For the special case where each non-zero value in $x^*$ is from $\{-\frac{1}{\sqrt{k}}, \frac{1}{\sqrt{k}}\}$, we have the following corollary:

**Corollary 5.2.** *Suppose $\mathbf{x}^*$ is $k$-sparse with non-zero elements $\pm\frac{1}{\sqrt{k}}$. If the number of measurements $m > c\left(k^2\log\frac{n}{\delta} + k\log^2 k + k\log\frac{1}{\epsilon}\right)$, then Algorithm 3 will recover $x^*$ up to accuracy $\epsilon$ with probability greater than $1 - \delta$.*

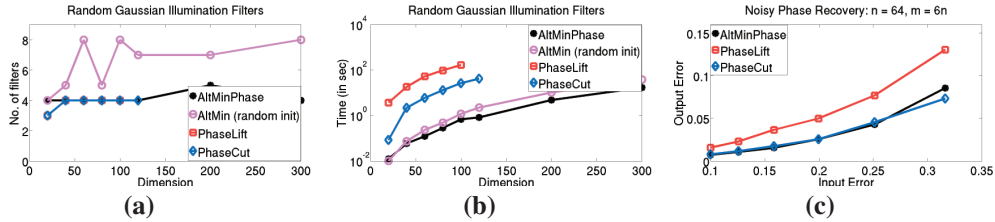

Figure 2: **(a) & (b)**: Sample and time complexity for successful recovery using random Gaussian illumination filters. Similar to Figure 1, we observe that AltMinPhase has similar number of filters ($J$) as PhaseLift and PhaseCut, but is computationally much more efficient. We also see that Alt-MinPhase performs better than AltMin (randominit). **(c)**: Recovery error $\|\mathbf{x} - \mathbf{x}^*\|_2$ incurred by various methods with increasing amount of noise ($\sigma$). AltMinPhase and PhaseCut perform comparably while PhaseLift incurs significantly larger error.

## 6  Experiments

In this section, we present experimental evaluation of AltMinPhase (Algorithm 1) and compare its performance with the SDP based methods PhaseLift [6] and PhaseCut [28]. We also empirically demonstrate the advantage of our initialization procedure over random initialization (denoted by **AltMin (random init)**), which has thus far been considered in the literature [13, 11, 28, 4]. **AltMin (random init)** is the same as AltMinPhase except that step 1 of Algorithm 1 is replaced with:$\mathbf{x^0} \leftarrow$ Uniformly random vector from the unit sphere.

We first choose $\mathbf{x}^*$ uniformly at random from the unit sphere. In the noiseless setting, a trial is said to **succeed** if the output $\mathbf{x}$ satisfies $\|\mathbf{x} - \mathbf{x}^*\|_2 < 10^{-2}$. For a given dimension, we do a linear search for smallest $m$ (number of samples) such that empirical success ratio over 20 runs is at least $0.8$. We implemented our methods in Matlab, while we obtained the code for PhaseLift and PhaseCut from the authors of [22] and [28] respectively.

We now present results from our experiments in three different settings.

**Independent Random Gaussian Measurements**: Each measurement vector $\mathbf{a_i}$ is generated from the standard complex Gaussian distribution. This measurement scheme was first suggested by [6] and till date, this is the only scheme with theoretical guarantees.

**Multiple Random Illumination Filters**: We now present our results for the setting where the measurements are obtained using multiple illumination filters; this setting was suggested by [4]. In particular, choose $J$ vectors $\mathbf{z}^{(1)}, \cdots, \mathbf{z}^{(J)}$ and compute the following discrete Fourier transforms:

$$\widehat{\mathbf{x}}^{(\mathbf{u})} = \text{DFT}\left(\mathbf{x}^* \cdot * \mathbf{z}^{(\mathbf{u})}\right),$$

where $\cdot *$ denotes component-wise multiplication. Our measurements will then be the magnitudes of components of the vectors $\widehat{\mathbf{x}}^{(1)}, \cdots, \widehat{\mathbf{x}}^{(\mathbf{J})}$. The above measurement scheme can be implemented by modulating the light beam or by the use of masks; see [4] for more details.

We again perform the same experiments as in the previous setting. Figures 2 (a) and (b) present the results. We again see that the measurement complexity of AltMinPhase is similar to that of PhaseCut and PhaseLift, but AltMinPhase is orders of magnitude faster than PhaseLift and PhaseCut.

**Noisy Phase Retrieval**: Finally, we study our method in the following noisy measurement scheme:

$$y_i = |\langle \mathbf{a}_i, \mathbf{x}^* + w_i \rangle| \qquad \text{for } i = 1, \dots, m, \tag{5}$$

where $w_i$ is the noise in the $i$-th measurement and is sampled from $\mathcal{N}(0, \sigma^2)$. We fix $n = 64$ and $m = 6n$. We then vary the amount of noise added $\sigma$ and measure the $\ell_2$ error in recovery, i.e., $\|\mathbf{x} - \mathbf{x}^*\|_2$, where $\mathbf{x}$ is the recovered vector. Figure 2(c) compares the performance of various methods with varying amount of noise. We observe that our method outperforms PhaseLift and has similar recovery error as PhaseCut.

### Acknowledgments

S. Sanghavi would like to acknowledge support from NSF grants 0954059, 1302435, ARO grant W911NF-11-1-0265 and a DTRA YIP award.

## Footnotes

[1]Our results also cover the real case, i.e. where all quantities are real.

[2] $z$ is standard complex Gaussian if $z = z_1 + i z_2$ where $z_1$ and $z_2$ are independent standard normal random variables.

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
