[Supplementary Material · phasesensing_supp.pdf]

# A  Proofs for Section 4

## A.1  Proof of the Initialization Step

*Proof of Theorem 4.1.* Recall that $\mathbf{x^0}$ is the top singular vector of $\mathbf{S} = \frac{1}{n}\sum_\ell |\mathbf{a}_\ell^T \mathbf{x}^*|^2 \mathbf{a}_\ell \mathbf{a}_\ell^T$. As $\mathbf{a}_\ell$ are rotationally invariant random variables, wlog, we can assume that $\mathbf{x}^* = \mathbf{e_1}$ where $\mathbf{e_1}$ is the first canonical basis vector. Also note that $\mathbb{E}\left[|\langle \mathbf{a}, \mathbf{e_1}\rangle|^2 \mathbf{a}\mathbf{a}^T\right] = \mathbf{D}$, where $\mathbf{D}$ is a diagonal matrix with $D_{11} = \mathbb{E}_{a\sim\mathcal{N}_C(0,1)}[|a|^4] = 8$ and $D_{ii} = \mathbb{E}_{a\sim\mathcal{N}_C(0,1), b\sim\mathcal{N}_C(0,1)}[|a|^2|b|^2] = 1, \forall i > 1$.

We break our proof of the theorem into two steps:
**(1)**: Show that, with probability $> 1 - \frac{4}{m^2}$: $\|\mathbf{S} - \mathbf{D}\|_2 < c/4$.
**(2)**: Use (1) to prove the theorem.

**Proof of Step (2)**: We have $\langle \mathbf{x^0}, \mathbf{S}\mathbf{x^0}\rangle \leq c/4 + 3\left((\mathbf{x^0})^T \mathbf{e_1}\right)^2 + \sum_{i=2}^n (\mathbf{x^0}_i)^2 = c/4 + 2\left((\mathbf{x^0})^T \mathbf{e_1}\right)^2 + 1$. On the other hand, since $\mathbf{x^0}$ is the top singular value of $\mathbf{S}$, by using triangle inequality, we have $\langle \mathbf{x^0}, \mathbf{S}\mathbf{x^0}\rangle > 3 - c/4$. Hence, $\langle \mathbf{x^0}, \mathbf{e_1}\rangle^2 > 1 - c/2$. This yields $\|\mathbf{x^0} - \mathbf{x}^*\|_2^2 = 2 - 2\langle \mathbf{x^0}, \mathbf{e_1}\rangle^2 < c$.

**Proof of Step (1)**: We now complete our proof by proving (1). To this end, we use the following matrix concentration result from [26]:

**Theorem A.1** (Theorem 1.5 of [26]). *Consider a finite sequence $\mathbf{X_i}$ of self-adjoint independent random matrices with dimensions $n \times n$. Assume that $\mathbb{E}[\mathbf{X_i}] = 0$ and $\|\mathbf{X_i}\|_2 \leq R, \forall i$, almost surely. Let $\sigma^2 := \|\sum_i \mathbb{E}[\mathbf{X_i}]\|_2$. Then the following holds $\forall \nu \geq 0$:*

$$P\left(\|\frac{1}{m}\sum_{i=1}^m \mathbf{X_i}\|_2 \geq \nu\right) \leq 2n \exp\left(\frac{-m^2\nu^2}{\sigma^2 + Rm\nu/3}\right).$$

Note that Theorem A.1 assumes $\max_\ell |a_{1\ell}|^2 \|\mathbf{a}_\ell\|^2$ to be bounded, where $a_{1\ell}$ is the first component of $\mathbf{a}_\ell$. However, $\mathbf{a}_\ell$ is a normal random variable and hence can be unbounded. We address this issue by observing that probability that $\Pr(\|\mathbf{a}_\ell\|^2 \geq 2n \ OR \ |a_{1\ell}|^2 \geq 2\log m) \leq 2\exp(-n/2) + \frac{1}{m^2}$. Hence, for large enough $n, \widehat{c}$ and $m > \widehat{c}n$, w.p. $1 - \frac{3}{m^2}$,

$$\max_\ell |a_{1\ell}|^2 \|\mathbf{a}_\ell\|^2 \leq 4n\log(m). \tag{6}$$

Now, consider truncated random variable $\widetilde{\mathbf{a}}_\ell$ s.t. $\widetilde{\mathbf{a}}_\ell = \mathbf{a}_\ell \ if \ |a_{1\ell}|^2 \leq 2\log(m)\&\|\mathbf{a}_\ell\|^2 \leq 2n$ and $\widetilde{\mathbf{a}}_\ell = 0$ otherwise. Now, note that $\widetilde{\mathbf{a}}_\ell$ is symmetric around origin and also $\mathbb{E}[\widetilde{a}_{i\ell}\widetilde{a}_{j\ell}] = 0, \forall i \neq j$. Also, $\mathbb{E}[|\widetilde{a}_{i\ell}|^2] \leq 1$. Hence, $\|\mathbb{E}[|\widetilde{a}_{1\ell}|^2 \|\widetilde{\mathbf{a}}_\ell\|^2 \widetilde{\mathbf{a}}_\ell \widetilde{\mathbf{a}}_\ell^\dagger]\|_2 \leq 4n\log(m)$. Now, applying Theorem A.1 given above, we get (w.p. $\geq 1 - 1/m^2$)

$$\|\frac{1}{m}\sum_\ell |\widetilde{a}_{1\ell}|^2 \widetilde{\mathbf{a}}_\ell \widetilde{\mathbf{a}}_\ell^\dagger - \mathbb{E}[|\widetilde{a}_{1\ell}|^2 \widetilde{\mathbf{a}}_\ell \widetilde{\mathbf{a}}_\ell^\dagger]\|_2 \leq \frac{4n\log^{3/2}(m)}{\sqrt{m}}.$$

Furthermore, $\mathbf{a}_\ell = \widetilde{\mathbf{a}}_\ell$ with probability larger than $1 - \frac{3}{m^2}$. Hence, w.p. $\geq 1 - \frac{4}{m^2}$:

$$\|S - \mathbb{E}[|\widetilde{\mathbf{a}}_\ell^1|^2 \widetilde{\mathbf{a}}_\ell \widetilde{\mathbf{a}}_\ell^\dagger]\|_2 \leq \frac{4n\log^{3/2}(m)}{\sqrt{m}}.$$

Now, the remaining task is to show that $\|\mathbb{E}[|\widetilde{\mathbf{a}}_\ell^1|^2 \widetilde{\mathbf{a}}_\ell \widetilde{\mathbf{a}}_\ell^\dagger] - \mathbb{E}[|\mathbf{a}_\ell^1|^2 \mathbf{a}_\ell \mathbf{a}_\ell^\dagger]\|_2 \leq \frac{1}{m}$. This follows easily by observing that $\mathbb{E}[\widetilde{\mathbf{a}}_\ell^i \widetilde{\mathbf{a}}_\ell^j] = 0$ and by bounding $\mathbb{E}[|\widetilde{\mathbf{a}}_\ell^1|^2 |\widetilde{\mathbf{a}}_\ell^i|^2 - |\mathbf{a}_\ell^1|^2 |\mathbf{a}_\ell^i|^2 \leq 1/m$ by using a simple second and fourth moment calculations for the normal distribution.

$\square$

## A.2  Proof of per step reduction in error

In all the lemmas in this section, $\delta$ is a small numerical constant (can be taken to be 0.01).

**Lemma A.2.** *Assume the hypothesis of Theorem 4.2 and let* $\mathbf{x}^+$ *be as defined in (3). Then, there exists an absolute numerical constant $c$ such that the following holds (w.p. $\geq 1 - \frac{\eta}{4}$):*
$$\left\| \left( \mathbf{A}\mathbf{A}^T \right)^{-1} \mathbf{A} \left( \mathbf{D} - \mathbf{I} \right) \mathbf{A}^T \mathbf{x}^* \right\|_2 < c\text{dist}\left( \mathbf{x}^*, \mathbf{x} \right).$$

*Proof.* Using (4) and the fact that $\|\mathbf{x}^*\|_2 = 1$, $\mathbf{x}^{*T}\mathbf{x}^+ = 1 + \mathbf{x}^{*T} \left( \mathbf{A}\mathbf{A}^T \right)^{-1} \mathbf{A} \left( \mathbf{D} - \mathbf{I} \right) \mathbf{A}^T \mathbf{x}^*$. That is, $|\mathbf{x}^{*T}\mathbf{x}^+| \geq 1 - \| \left( \frac{1}{2m}\mathbf{A}\mathbf{A}^T \right)^{-1} \|_2 \frac{1}{\sqrt{2m}}A\|_2 \frac{1}{\sqrt{2m}} \left( \mathbf{D} - \mathbf{I} \right) \mathbf{A}^T \mathbf{x}^*\|_2$. Now, using standard bounds on the singular values of Gaussian matrices [27] and assuming $m > \widehat{c}\log\frac{1}{\eta}n$, we have (w.p. $\geq 1 - \frac{\eta}{4}$): $\| \left( \frac{1}{2m}\mathbf{A}\mathbf{A}^T \right)^{-1} \|_2 \leq 1/(1 - 2/\sqrt{\widehat{c}})^2$ and $\|\mathbf{A}\|_2 \leq 1 + 2/\sqrt{\widehat{c}}$. Note that both the quantities can be bounded by constants that are close to 1 by selecting a large enough $\widehat{c}$. Also note that $\frac{1}{2m}\mathbf{A}\mathbf{A}^T$ converges to $\mathbf{I}$ (the identity matrix), or equivalently $\frac{1}{m}\mathbf{A}\mathbf{A}^T$ converges to $2\mathbf{I}$ since the elements of $A$ are standard normal complex random variables and not standard normal real random variables.

The key challenge now is to bound $\left\| \left( \mathbf{D} - \mathbf{I} \right) \mathbf{A}^T \mathbf{x}^* \right\|_2$ by $c\sqrt{m}\text{dist}\left( \mathbf{x}^*, \mathbf{x}^t \right)$ for a global constant $c > 0$. Note that since (4) is invariant with respect to $\left\| \mathbf{x}^t \right\|_2$, we can assume that $\left\| \mathbf{x}^t \right\|_2 = 1$. Note further that, since the distribution of $\mathbf{A}$ is rotationally invariant and is independent of $\mathbf{x}^*$ and $\mathbf{x}^t$, wlog, we can assume that $\mathbf{x}^* = \mathbf{e}_1$ and $\mathbf{x}^t = \alpha\mathbf{e_1} + \sqrt{1 - \alpha^2}\mathbf{e_2}$, where $\alpha = \langle \mathbf{x}^t, \mathbf{x}^* \rangle$.

Hence, $\left\| \left( \mathbf{D} - \mathbf{I} \right) \mathbf{A}^T \mathbf{e_1} \right\|_2^2 = \sum_{l=1}^m |a_{1l}|^2 \left| \text{Ph}\left( \left( \alpha\bar{a}_{1l} + \sqrt{1 - \alpha^2}\bar{a}_{2l} \right) a_{1l} \right) - 1 \right|^2 = \sum_{l=1}^m U_\ell$, where $U_l$ is given by,

$$U_l \overset{\text{def}}{=} |a_{1l}|^2 \left| \text{Ph}\left( \left( \alpha\bar{a}_{1l} + \sqrt{1 - \alpha^2}\bar{a}_{2l} \right) a_{1l} \right) - 1 \right|^2. \tag{7}$$

Using Lemma A.3 finishes the proof. $\square$

The following lemma, Lemma A.3 shows that if $U_\ell$ are as defined in Lemma A.2 then, the sum of $U_\ell, 1 \leq \ell \leq m$ concentrates well around $\mathbb{E}\left[ U_\ell \right]$ and also $\mathbb{E}\left[ U_\ell \right] \leq c\sqrt{m}\text{dist}\left( \mathbf{x}^*, \mathbf{x}^t \right)$. The proof of Lemma A.3 requires careful analysis as it provides tail bound and expectation bound of a random variable that is a product of correlated sub-exponential complex random variables.

**Lemma A.3.** *Assume the hypothesis of Lemma A.2. Let $U_\ell$ be as defined in (7) and let each $a_{1l}, a_{2l}, \forall 1 \leq l \leq m$ be sampled from standard normal distribution for complex numbers. Then, with probability greater than $1 - \frac{\eta}{4}$, we have: $\sum_{l=1}^m U_l \leq c^2 m(1 - \alpha^2)$, for a global constant $c > 0$.*

*Proof of Lemma A.3.* We first estimate $\mathbb{P}\left[ U_l > t \right]$ so as to:

1. Calculate $\mathbb{E}\left[ U_l \right]$ and,

2. Show that $U_l$ is a subexponential random variable and use that fact to derive concentration bounds.

Now, $\mathbb{P}\left[ U_l > t \right] = \int_{\frac{\sqrt{t}}{2}}^{\infty} p_{|a_{1l}|}(s)\mathbb{P}\left[ W_l > \frac{\sqrt{t}}{s} \Big| |a_{1l}| = s \right] ds$, where,

$$W_l \overset{\text{def}}{=} \left| \text{Ph}\left( \left( \alpha\bar{a}_{1l} + \sqrt{1 - \alpha^2}\bar{a}_{2l} \right) a_{1l} \right) - 1 \right|.$$

$$\mathbb{P}\left[W_l > \frac{\sqrt{t}}{s}\Big||a_{1l}| = s\right] = \mathbb{P}\left[\left|\text{Ph}\left(\left(\alpha\bar{a}_{1l} + \sqrt{1-\alpha^2}\bar{a}_{2l}\right)a_{1l}\right) - 1\right| > \frac{\sqrt{t}}{s}\Big||a_{1l}| = s\right]$$

$$= \mathbb{P}\left[\left|\text{Ph}\left(1 + \frac{\sqrt{1-\alpha^2}\bar{a}_{2l}}{\alpha\bar{a}_{1l}}\right) - 1\right| > \frac{\sqrt{t}}{s}\Big||a_{1l}| = s\right]$$

$$\overset{(\zeta_1)}{\leq} \mathbb{P}\left[\frac{\sqrt{1-\alpha^2}\,|a_{2l}|}{\alpha\,|a_{2l}|} > \frac{c\sqrt{t}}{s}\Big||a_{1l}| = s\right]$$

$$= \mathbb{P}\left[|a_{2l}| > \frac{c\alpha\sqrt{t}}{\sqrt{1-\alpha^2}}\right]$$

$$\overset{(\zeta_2)}{\leq} \exp\left(1 - \frac{c\alpha^2 t}{1-\alpha^2}\right),$$

where $(\zeta_1)$ follows from Lemma A.7 and $(\zeta_2)$ follows from the fact that $a_{2l}$ is a sub-gaussian random variable. So we have:

$$\mathbb{P}\left[U_l > t\right] \leq \int_{\frac{\sqrt{t}}{2}}^{\infty} \exp\left(1 - \frac{c\alpha^2 t}{1-\alpha^2}\right)ds = \exp\left(1 - \frac{c\alpha^2 t}{1-\alpha^2}\right)\int_{\frac{\sqrt{t}}{2}}^{\infty} se^{-\frac{s^2}{2}}ds = \exp\left(1 - \frac{ct}{1-\alpha^2}\right). \tag{8}$$

Using this, we have the following bound on the expected value of $U_l$:

$$\mathbb{E}[U_l] = \int_0^{\infty}\mathbb{P}[U_l > t]\,dt \leq \int_0^{\infty}\exp\left(1 - \frac{ct}{1-\alpha^2}\right)dt \leq c\left(1-\alpha^2\right). \tag{9}$$

From (8), we see that $U_l$ is a subexponential random variable with parameter $c\left(1-\alpha^2\right)$. Using Proposition 5.16 from [27], we obtain:

$$\mathbb{P}\left[\left|\sum_{l=1}^m U_l - \mathbb{E}[U_l]\right| > \delta m\left(1-\alpha^2\right)\right] \leq 2\exp\left(-\min\left(\frac{c\delta^2 m^2\left(1-\alpha^2\right)^2}{\left(1-\alpha^2\right)^2 m}, \frac{c\delta m\left(1-\alpha^2\right)}{1-\alpha^2}\right)\right)$$

$$\leq 2\exp\left(-c\delta^2 m\right) \leq \frac{\eta}{4}.$$

So, with probability greater than $1 - \frac{\eta}{4}$, we have:

$$\sum_{l=1}^m U_l \leq c^2 m(1-\alpha^2).$$

This proves the lemma. $\square$

**Lemma A.4.** *Assume the hypothesis of Theorem 4.2 and let $\mathbf{x}^+$ be as defined in (3). Then, $\forall \mathbf{z}$ s.t. $\langle \mathbf{z}, \mathbf{x}^* \rangle = 0$, the following holds (w.p. $\geq 1 - \frac{\eta}{4}e^{-n}$): $|\langle \mathbf{z}, \mathbf{x}^+ \rangle| \leq \frac{5}{9}\text{dist}(\mathbf{x}^*, \mathbf{x})$.*

*Proof.* Fix $\mathbf{z}$ such that $\langle \mathbf{z}, \mathbf{x}^* \rangle = 0$. Since the distribution of $A$ is rotationally invariant, wlog we can assume that: a) $\mathbf{x}^* = \mathbf{e_1}$, b) $\mathbf{x} = \alpha\mathbf{e_1} + \sqrt{1-\alpha^2}\mathbf{e_2}$ where $\alpha \in \mathbb{R}$ and $\alpha \geq 0$ and c) $\mathbf{z} = \beta\mathbf{e_2} + \sqrt{1-|\beta|^2}\mathbf{e_3}$ for some $\beta \in \mathbb{C}$. Note that we first prove the lemma for a *fixed* $\mathbf{z}$ and then using union bound, we obtain the result $\forall \mathbf{z} \in \mathbb{C}^n$. We have:

$$|\langle \mathbf{z}, \mathbf{x}^+ \rangle| \leq |\beta|\,|\langle \mathbf{e_2}, \mathbf{x}^+ \rangle| + \sqrt{1-|\beta|^2}|\langle \mathbf{e_3}, \mathbf{x}^+ \rangle|. \tag{10}$$

Now,

$$\left|\mathbf{e_2}^T\mathbf{x}^+\right| = \left|\mathbf{e_2}^T\left(\mathbf{A}\mathbf{A}^T\right)^{-1}\mathbf{A}\left(\mathbf{D} - \mathbf{I}\right)\mathbf{A}^T\mathbf{e_1}\right|$$

$$\leq \frac{1}{2m}\left|\mathbf{e_2}^T\left(\left(\frac{1}{2m}\mathbf{A}\mathbf{A}^T\right)^{-1} - \mathbf{I}\right)\mathbf{A}\left(\mathbf{D} - \mathbf{I}\right)\mathbf{A}^T\mathbf{e_1}\right| + \frac{1}{2m}\left|\mathbf{e_2}^T\mathbf{A}\left(\mathbf{D} - \mathbf{I}\right)\mathbf{A}^T\mathbf{e_1}\right|$$

$$\leq \frac{1}{2m}\left\|\left(\frac{1}{2m}\mathbf{A}\mathbf{A}^T\right)^{-1} - \mathbf{I}\right\|_2\|\mathbf{A}\|_2\left\|\left(\mathbf{D} - \mathbf{I}\right)\mathbf{A}^T\mathbf{e_1}\right\|_2 + \frac{1}{2m}\left|\mathbf{e_2}^T\mathbf{A}\left(\mathbf{D} - \mathbf{I}\right)\mathbf{A}^T\mathbf{e_1}\right|,$$

$$\leq \frac{4c}{\sqrt{\hat{c}}}\text{dist}\left(\mathbf{x^t}, \mathbf{x}^*\right) + \frac{1}{2m}\left|\mathbf{e_2}^T\mathbf{A}\left(\mathbf{D} - \mathbf{I}\right)\mathbf{A}^T\mathbf{e_1}\right|, \tag{11}$$

where the last inequality follows from the proof of Lemma A.2.

Similarly,

$$
\begin{aligned}
\left|\mathbf{e_3}^T\mathbf{x}^+\right| &= \left|\mathbf{e_3}^T\left(\mathbf{AA}^T\right)^{-1}\mathbf{A}\left(\mathbf{D}-\mathbf{I}\right)\mathbf{A}^T\mathbf{e_1}\right| \\
&\leq \frac{1}{2m}\left|\mathbf{e_3}^T\left(\left(\frac{1}{2m}\mathbf{AA}^T\right)^{-1}-\mathbf{I}\right)\mathbf{A}\left(\mathbf{D}-\mathbf{I}\right)\mathbf{A}^T\mathbf{e_1}\right| + \frac{1}{2m}\left|\mathbf{e_3}^T\mathbf{A}\left(\mathbf{D}-\mathbf{I}\right)\mathbf{A}^T\mathbf{e_1}\right| \\
&\leq \frac{1}{2m}\left\|\left(\frac{1}{2m}\mathbf{AA}^T\right)^{-1}-\mathbf{I}\right\|_2\left\|\mathbf{A}\right\|_2\left\|\left(\mathbf{D}-\mathbf{I}\right)\mathbf{A}^T\mathbf{e_1}\right\|_2 + \frac{1}{2m}\left|\mathbf{e_3}^T\mathbf{A}\left(\mathbf{D}-\mathbf{I}\right)\mathbf{A}^T\mathbf{e_1}\right| \\
&\leq \frac{4c}{\sqrt{\widehat{c}}}\mathrm{dist}\left(\mathbf{x^t},\mathbf{x}^*\right) + \frac{1}{2m}\left|\mathbf{e_3}^T\mathbf{A}\left(\mathbf{D}-\mathbf{I}\right)\mathbf{A}^T\mathbf{e_1}\right|,
\end{aligned}
\tag{12}
$$

Again, the last inequality follows from the proof of Lemma A.2. The lemma now follows by using (10), (11), (12) along with Lemmas A.5 and A.6. $\qquad\square$

**Lemma A.5.** *Assume the hypothesis of Theorem 4.2 and the notation therein. Then,*

$$
\left|\mathbf{e_2}^T\mathbf{A}\left(\mathbf{D}-\mathbf{I}\right)\mathbf{A}^T\mathbf{e_1}\right| \leq \frac{100}{99}m\sqrt{1-\alpha^2},
$$

*with probability greater than $1-\frac{\eta}{10}e^{-n}$.*

*Proof.* We have:

$$
\begin{aligned}
\mathbf{e_2}^T\mathbf{A}\left(\mathbf{D}-\mathbf{I}\right)\mathbf{A}^T\mathbf{e_1} &= \sum_{l=1}^m \bar{a}_{1l}a_{2l}\left(\mathrm{Ph}\left(\left(\alpha\bar{a}_{1l}+\sqrt{1-\alpha^2}\bar{a}_{2l}\right)a_{1l}\right)-1\right) \\
&= \sum_{l=1}^m |a_{1l}|\,a'_{2l}\left(\mathrm{Ph}\left(\alpha\,|a_{1l}|+\sqrt{1-\alpha^2}\overline{a'_{2l}}\right)-1\right),
\end{aligned}
$$

where $a'_{2l} \overset{\text{def}}{=} a_{2l}\mathrm{Ph}\left(\bar{a}_{1l}\right)$ is identically distributed to $a_{2l}$ and is independent of $|a_{1l}|$. Define the random variable $U_l$ as:

$$
U_l \overset{\text{def}}{=} |a_{1l}|\,a'_{2l}\left(\mathrm{Ph}\left(1+\frac{\sqrt{1-\alpha^2}a'_{2l}}{\alpha\,|a_{1l}|}\right)-1\right).
$$

Similar to Lemma A.2, we will calculate $\mathbb{P}\left[U_l > t\right]$ to show that $U_l$ is subexponential and use it to derive concentration bounds. However, using the above estimate to bound $\mathbb{E}\left[U_l\right]$ will result in a weak bound that we will not be able to use. Lemma 4.3 bounds $\mathbb{E}\left[U_l\right]$ using a different technique carefully.

$$
\begin{aligned}
\mathbb{P}\left[|U_l| > t\right] &\leq \mathbb{P}\left[|a_{1l}|\,|a'_{2l}|\frac{c\sqrt{1-\alpha^2}\,|a'_{2l}|}{\alpha\,|a_{1l}|} > t\right] \\
&= \mathbb{P}\left[|a'_{2l}|^2 > \frac{c\alpha t}{\sqrt{1-\alpha^2}}\right] \leq \exp\left(1-\frac{c\alpha t}{\sqrt{1-\alpha^2}}\right),
\end{aligned}
$$

where the last step follows from the fact that $a'_{2l}$ is a subgaussian random variable and hence $|a'_{2l}|^2$ is a subexponential random variable. Using Proposition 5.16 from [27], we obtain:

$$
\begin{aligned}
\mathbb{P}\left[\left|\sum_{l=1}^m U_l - \mathbb{E}\left[U_l\right]\right| > \delta m\sqrt{1-\alpha^2}\right] &\leq 2\exp\left(-\min\left(\frac{c\delta^2 m^2\left(1-\alpha^2\right)}{\left(1-\alpha^2\right)m},\frac{c\delta m\sqrt{1-\alpha^2}}{\sqrt{1-\alpha^2}}\right)\right) \\
&\leq 2\exp\left(-c\delta^2 m\right) \leq \frac{\eta}{10}\exp\left(-n\right).
\end{aligned}
$$

Using Lemma 4.3, we obtain:

$$
\left|\mathbf{e_2}^T\mathbf{A}\left(\mathbf{D}-\mathbf{I}\right)\mathbf{A}^T\mathbf{e_1}\right| = \left|\sum_{l=1}^m U_l\right| \leq \left(1+\delta\right)m\sqrt{1-\alpha^2},
$$

with probability greater than $1-\frac{\eta}{10}\exp(-n)$. This proves the lemma. $\qquad\square$

*Proof of Lemma 4.3.* Let $w_2 = |w_2| e^{i\theta}$. Then $|w_1|, |w_2|$ and $\theta$ are all independent random variables. $\theta$ is a uniform random variable over $[-\pi, \pi]$ and $|w_1|$ and $|w_2|$ are identically distributed with probability distribution function:

$$p(x) = x \exp\left(-\frac{x^2}{2}\right) \mathbb{1}_{\{x \geq 0\}}.$$

We have:

$$\mathbb{E}\left[U\right] = \mathbb{E}\left[|w_1||w_2| e^{i\theta} \left(\mathrm{Ph}\left(1 + \frac{\sqrt{1-\alpha^2}|w_2| e^{-i\theta}}{\alpha|w_1|}\right) - 1\right)\right]$$

$$= \mathbb{E}\left[|w_1||w_2| \mathbb{E}\left[e^{i\theta}\left(\mathrm{Ph}\left(1 + \frac{\sqrt{1-\alpha^2}|w_2| e^{-i\theta}}{\alpha|w_1|}\right) - 1\right)\Big||w_1|, |w_2|\right]\right]$$

Let $\beta \stackrel{\text{def}}{=} \frac{\sqrt{1-\alpha^2}|w_2|}{\alpha|w_1|}$. We will first calculate $\mathbb{E}\left[e^{i\theta}\mathrm{Ph}\left(1 + \beta e^{-i\theta}\right)\big||w_1|, |w_2|\right]$. Note that the above expectation is taken only over the randomness in $\theta$. For simplicity of notation, we will drop the conditioning variables, and calculate the above expectation in terms of $\beta$.

$$e^{i\theta}\mathrm{Ph}\left(1 + \beta e^{-i\theta}\right) = (\cos\theta + i\sin\theta) \frac{1 + \beta\cos\theta - i\beta\sin\theta}{\left[(1+\beta\cos\theta)^2 + \beta^2\sin^2\theta\right]^{\frac{1}{2}}}$$

$$= \frac{\cos\theta + \beta + i\sin\theta}{(1 + \beta^2 + 2\beta\cos\theta)^{\frac{1}{2}}}.$$

We will first calculate the imaginary part of the above expectation:

$$\mathrm{Im}\left(\mathbb{E}\left[e^{i\theta}\mathrm{Ph}\left(1 + \beta e^{-i\theta}\right)\right]\right) = \mathbb{E}\left[\frac{\sin\theta}{(1 + \beta^2 + 2\beta\cos\theta)^{\frac{1}{2}}}\right] = 0, \tag{13}$$

where the last step follows because we are taking the expectation of an odd function. Focusing on the real part, we let:

$$F(\beta) \stackrel{\text{def}}{=} \mathbb{E}\left[\frac{\cos\theta + \beta}{(1 + \beta^2 + 2\beta\cos\theta)^{\frac{1}{2}}}\right]$$

$$= \frac{1}{2\pi}\int_{-\pi}^{\pi} \frac{\cos\theta + \beta}{(1 + \beta^2 + 2\beta\cos\theta)^{\frac{1}{2}}} d\theta.$$

Note that $F(\beta) : \mathbb{R} \to \mathbb{R}$ and $F(0) = 0$. We will show that there is a small absolute numerical constant $\gamma$ (depending on $\delta$) such that:

$$0 < \beta < \gamma \Rightarrow |F(\beta)| \leq (\frac{1}{2} + \delta)\beta. \tag{14}$$

We show this by calculating $F'(0)$ and using the continuity of $F'(\beta)$ at $\beta = 0$. We first calculate $F'(\beta)$ as follows:

$$F'(\beta) = \frac{1}{2\pi}\int_{-\pi}^{\pi} \frac{1}{(1 + \beta^2 + 2\beta\cos\theta)^{\frac{1}{2}}} - \frac{(\cos\theta + \beta)(\beta + \cos\theta)}{(1 + \beta^2 + 2\beta\cos\theta)^{\frac{3}{2}}} d\theta$$

$$= \frac{1}{2\pi}\int_{-\pi}^{\pi} \frac{\sin^2\theta}{(1 + \beta^2 + 2\beta\cos\theta)^{\frac{3}{2}}} d\theta$$

From the above, we see that $F'(0) = \frac{1}{2}$ and (14) then follows from the continuity of $F'(\beta)$ at $\beta = 0$. Getting back to the expected value of $U$, we have:

$$|\mathbb{E}[U]| = \left| \mathbb{E}\left[ |w_1|\,|w_2|\, F\left( \frac{\sqrt{1-\alpha^2}\,|w_2|}{\alpha\,|w_1|} \right) \mathbb{1}_{\left\{ \frac{\sqrt{1-\alpha^2}|w_2|}{\alpha|w_1|} < \gamma \right\}} \right] \right.$$
$$\left. + \mathbb{E}\left[ |w_1|\,|w_2|\, F\left( \frac{\sqrt{1-\alpha^2}\,|w_2|}{\alpha\,|w_1|} \right) \mathbb{1}_{\left\{ \frac{\sqrt{1-\alpha^2}|w_2|}{\alpha|w_1|} \geq \gamma \right\}} \right] \right|$$

$$= \left| \mathbb{E}\left[ |w_1|\,|w_2|\, F\left( \frac{\sqrt{1-\alpha^2}\,|w_2|}{\alpha\,|w_1|} \right) \mathbb{1}_{\left\{ \frac{\sqrt{1-\alpha^2}|w_2|}{\alpha|w_1|} < \gamma \right\}} \right] \right|$$
$$+ \left| \mathbb{E}\left[ |w_1|\,|w_2|\, F\left( \frac{\sqrt{1-\alpha^2}\,|w_2|}{\alpha\,|w_1|} \right) \mathbb{1}_{\left\{ \frac{\sqrt{1-\alpha^2}|w_2|}{\alpha|w_1|} \geq \gamma \right\}} \right] \right|$$

$$\overset{(\zeta_1)}{\leq} \left( \frac{1}{2} + \delta \right) \mathbb{E}\left[ |w_1|\,|w_2|\, \frac{\sqrt{1-\alpha^2}\,|w_2|}{\alpha\,|w_1|} \right] + \mathbb{E}\left[ |w_1|\,|w_2|\, \mathbb{1}_{\left\{ \frac{\sqrt{1-\alpha^2}|w_2|}{\alpha|w_1|} \geq \gamma \right\}} \right],$$

$$= \left( \frac{1}{2} + \delta \right) \left( \frac{\sqrt{1-\alpha^2}}{\alpha} \right) \mathbb{E}\left[ |w_2|^2 \right] + \mathbb{E}\left[ |w_1|\,|w_2|\, \mathbb{1}_{\left\{ \frac{\sqrt{1-\alpha^2}|w_2|}{\alpha|w_1|} \geq \gamma \right\}} \right],$$

$$\overset{(\zeta_2)}{=} (1 + 2\delta) \left( \frac{\sqrt{1-\alpha^2}}{\alpha} \right) + \mathbb{E}\left[ |w_1|\,|w_2|\, \mathbb{1}_{\left\{ \frac{\sqrt{1-\alpha^2}|w_2|}{\alpha|w_1|} \geq \gamma \right\}} \right], \tag{15}$$

where $(\zeta_1)$ follows from (14) and the fact that $|F(\beta)| \leq 1$ for every $\beta$ and $(\zeta_2)$ follows from the fact that $\mathbb{E}\left[ |z_2|^2 \right] = 2$. We will now bound the second term in the above inequality. We start with the following integral:

$$\int_t^\infty s^2 e^{-\frac{s^2}{2}}\, ds = -\int_t^\infty s\, d\left( e^{-\frac{s^2}{2}} \right)$$
$$= t e^{-\frac{t^2}{2}} + \int_t^\infty e^{-\frac{s^2}{2}}\, ds \leq (t+e) e^{-\frac{t^2}{c}}, \tag{16}$$

where $c$ is some constant. The last step follows from standard bounds on the tail probabilities of gaussian random variables. We now bound the second term of (15) as follows:

$$\mathbb{E}\left[ |w_1|\,|w_2|\, \mathbb{1}_{\left\{ \frac{\sqrt{1-\alpha^2}|w_2|}{\alpha|w_1|} \geq \gamma \right\}} \right] = \int_0^\infty t^2 e^{-\frac{t^2}{2}} \int_{\frac{\alpha t}{\sqrt{1-\alpha^2}}}^\infty s^2 e^{-\frac{s^2}{2}}\, ds\, dt$$

$$\overset{(\zeta_1)}{\leq} \int_0^\infty t^2 e^{-\frac{t^2}{2}} \left( \frac{\alpha t}{\sqrt{1-\alpha^2}} + e \right) e^{-\frac{\alpha^2 t^2}{c(1-\alpha^2)}}\, dt$$

$$\leq \int_0^\infty \left( \frac{\alpha t^3}{\sqrt{1-\alpha^2}} + e t^2 \right) e^{-\frac{t^2}{c(1-\alpha^2)}}\, dt$$

$$= \frac{\alpha}{\sqrt{1-\alpha^2}} \int_0^\infty t^3 e^{-\frac{t^2}{c(1-\alpha^2)}}\, dt + e \int_0^\infty t^2 e^{-\frac{t^2}{c(1-\alpha^2)}}\, dt$$

$$\overset{(\zeta_2)}{\leq} c\left( 1-\alpha^2 \right)^{\frac{3}{2}} \overset{(\zeta_3)}{\leq} \delta\sqrt{1-\alpha^2}$$

where $(\zeta_1)$ follows from (16), $(\zeta_2)$ follows from the formulae for second and third absolute moments of gaussian random variables and $(\zeta_3)$ follows from the fact that $1 - \alpha^2 < \delta$. Plugging the above inequality in (15), we obtain:

$$|\mathbb{E}[U]| \leq (1 + 2\delta) \left( \frac{\sqrt{1-\alpha^2}}{\alpha} \right) + \delta\sqrt{1-\alpha^2} \leq (1 + 4\delta)\sqrt{1-\alpha^2},$$

where we used the fact that $\alpha \geq 1 - \frac{\delta}{2}$. This proves the lemma. $\qquad \square$

**Lemma A.6.** *Assume the hypothesis of Theorem 4.2 and the notation therein. Then,*

$$\left|\mathbf{e_3}^T \mathbf{A} \left(\mathbf{D} - \mathbf{I}\right) \mathbf{A}^T \mathbf{e_1}\right| \leq \delta m \sqrt{1 - \alpha^2},$$

*with probability greater than* $1 - \frac{\eta}{10} e^{-n}$.

*Proof.* The proof of this lemma is very similar to that of Lemma A.5. We have:

$$\mathbf{e_3}^T \mathbf{A} \left(\mathbf{D} - \mathbf{I}\right) \mathbf{A}^T \mathbf{e_1} = \sum_{l=1}^{m} \bar{a}_{1l} a_{3l} \left(\mathrm{Ph}\left(\left(\alpha \bar{a}_{1l} + \bar{a}_{2l}\sqrt{1 - \alpha^2}\bar{a}_{3l}\right) a_{1l}\right) - 1\right)$$

$$= \sum_{l=1}^{m} |a_{1l}| \, a'_{3l} \left(\mathrm{Ph}\left(\alpha |a_{1l}| + \overline{a'_{2l}}\sqrt{1 - \alpha^2}\right) - 1\right),$$

where $a'_{3l} \overset{\text{def}}{=} a_{3l}\mathrm{Ph}\left(\bar{a}_{1l}\right)$ is identically distributed to $a_{3l}$ and is independent of $|a_{1l}|$ and $a'_{2l}$. Define the random variable $U_l$ as:

$$U_l \overset{\text{def}}{=} |a_{1l}| \, a'_{3l} \left(\mathrm{Ph}\left(1 + \frac{\overline{a'_{2l}}\sqrt{1 - \alpha^2}}{\alpha |a_{1l}|}\right) - 1\right).$$

Since $a'_{3l}$ has mean zero and is independent of everything else, we have:

$$\mathbb{E}\left[U_l\right] = 0.$$

Similar to Lemma A.5, we will calculate $\mathbb{P}\left[U_l > t\right]$ to show that $U_l$ is subexponential and use it to derive concentration bounds.

$$\mathbb{P}\left[|U_l| > t\right] \leq \mathbb{P}\left[|a_{1l}| \, |a'_{3l}| \frac{c\sqrt{1 - \alpha^2} \, |a'_{2l}|}{\alpha |a_{1l}|} > t\right]$$

$$= \mathbb{P}\left[|a'_{2l} a'_{3l}| > \frac{c\alpha t}{\sqrt{1 - \alpha^2}}\right] \leq \exp\left(1 - \frac{c\alpha t}{\sqrt{1 - \alpha^2}}\right),$$

where the last step follows from the fact that $a'_{2l}$ and $a'_{3l}$ are independent subgaussian random variables and hence $|a'_{2l} a'_{3l}|$ is a subexponential random variable. Using Proposition 5.16 from [27], we obtain:

$$\mathbb{P}\left[\left|\sum_{l=1}^{m} U_l - \mathbb{E}\left[U_l\right]\right| > \delta m \sqrt{1 - \alpha^2}\right] \leq 2\exp\left(-\min\left(\frac{c\delta^2 m^2 \left(1 - \alpha^2\right)}{\left(1 - \alpha^2\right) m}, \frac{c\delta m\sqrt{1 - \alpha^2}}{\sqrt{1 - \alpha^2}}\right)\right)$$

$$\leq 2\exp\left(-c\delta^2 m\right) \leq \frac{\eta}{10}\exp\left(-n\right).$$

Hence, we have:

$$\left|\mathbf{e_3}^T \mathbf{A} \left(\mathbf{D} - \mathbf{I}\right) \mathbf{A}^T \mathbf{e_1}\right| = \left|\sum_{l=1}^{m} U_l\right| \leq \delta m \sqrt{1 - \alpha^2},$$

with probability greater than $1 - \frac{\eta}{10}\exp(-n)$. This proves the lemma. $\square$

**Lemma A.7.** *For every* $w \in \mathbb{C}$, *we have:*

$$\left|\mathrm{Ph}\left(1 + w\right) - 1\right| \leq 2\,|w|.$$

*Proof.* The proof is straight forward:

$$\left|\mathrm{Ph}\left(1 + w\right) - 1\right| \leq \left|\mathrm{Ph}\left(1 + w\right) - \left(1 + w\right)\right| + |w| = \left|1 - |1 + w|\right| + |w| \leq 2\,|w|.$$

$\square$

# B  Proofs for Section 5

*Proof of Lemma 5.1.* For every $j \in [n]$ and $i \in [m]$, consider the random variable $Z_{ij} \stackrel{\text{def}}{=} |a_{ij} y_i|$. We have the following:

- if $j \in S$, then

$$\mathbb{E}\left[Z_{ij}\right] = \frac{2}{\pi} \left( \sqrt{1 - \left(x_j^*\right)^2} + x_j^* \arcsin x_j^* \right)$$

$$\geq \frac{2}{\pi} \left( 1 - \frac{5}{6} \left(x_j^*\right)^2 - \frac{1}{6} \left(x_j^*\right)^4 + x_j^* \left( x_j^* + \frac{1}{6} \left(x_j^*\right)^3 \right) \right)$$

$$\geq \frac{2}{\pi} + \frac{1}{6} \left(x_{\min}^*\right)^2,$$

  where the first step follows from Corollary 3.1 in [17] and the second step follows from the Taylor series expansions of $\sqrt{1 - x^2}$ and $\arcsin(x)$,

- if $j \notin S$, then $\mathbb{E}\left[Z_{ij}\right] = \mathbb{E}\left[|a_{ij}|\right] \mathbb{E}\left[|y_i|\right] = \frac{2}{\pi}$ and finally,

- for every $j \in [n]$, $Z_{ij}$ is a sub-exponential random variable with parameter $c = O(1)$ (since it is a product of two standard normal random variables).

Using the hypothesis of the theorem about $m$, we have:

- for any $j \in S$, $\mathbb{P}\left[\frac{1}{m} \sum_{i=1}^{m} Z_{ij} - \left(\frac{2}{\pi} + \frac{1}{12} \left(x_{\min}^*\right)^2\right) < 0\right] \leq \exp\left(-c\left(x_{\min}^*\right)^4 m\right) \leq \delta n^{-c}$, and

- for any $j \notin S$, $\mathbb{P}\left[\frac{1}{m} \sum_{i=1}^{m} Z_{ij} - \left(\frac{2}{\pi} + \frac{1}{12} \left(x_{\min}^*\right)^2\right) > 0\right] \leq \exp\left(-c\left(x_{\min}^*\right)^4 m\right) \leq \delta n^{-c}$.

Applying a union bound to the above, we see that with probability greater than $1 - \delta$, there is a separation in the values of $\frac{1}{m} \sum_{i=1}^{m} Z_{ij}$ for $j \in S$ and $j \notin S$. This proves the theorem. $\quad\square$