[Reviews · NeurIPS 2013]

Submitted by Assigned_Reviewer_4

This paper presents a simple modification of the alternating minimization algorithm for phase retrieval
and analysis to prove its convergence. In phase retrieval the measurement model is based on a linear
system y = A x where the common term y (the measurements) is known up to the sign or phase,
A is given and one needs to find x.
The modification lies in the initialization. Prior alternating minimization algorithms used a random
guess for the reconstructed signal x. The proposed modification is to use the main singular vector
of a matrix built with a subset of columns of A and the corresponding entries of the vector y.
Then, at each iteration of the alternating minimization a new subset of the columns of A and the
corresponding entries of y is used until no columns are left.
The analysis shows that this algorithm can converge to the correct solution within a given
error with high probability when the measurements y_i are iid standard complex Gaussians random
variables. A second result applies to the case of phase retrieval problems where the unknown signal
is sparse. The authors propose to first identify the nonzero entries by a greedy algorithm and
then suggest to run the previous phase algorithm only on the nonzero entries.
They also characterize the probability that the greedy algorithm finds the correct support of the signal.

Quality
--
I have tried to verify the proofs in the paper, but it is not feasible to review 17 pages of proofs
in the supplementary material... this is more than 3 papers in one.
I have looked at some proofs, but I can't see how the authors do some steps.
For example, in lines 308-9 I could not verify the inequalities. Besides, the sentence
in line 311 does not make much sense. I also believe that the solution in C at lines 190-1
is missing y and the expression for S at line 199 does not match that in Algorithm 1 at line 1.
Probably a square of y is missing.

Even if all proofs were correct, what is missing to me is how practical this strategy is.
Essentially we have a linear system where the phase/sign of the measurements is lost.
The idea is to split this (non)linear system into a collection of (non)linear systems such that
each system has enough equations to yield a unique solution and then one can use each
system sequentially to arrive at the final solution.
The question is then: How many equations does one need to achieve a reasonable
convergence with reasonably high probability? To guarantee convergence with high
probability one needs a very small \eta in Thm 4.4 and this constraints the constant c
which in turn provides a lower bound for the number of measurements m.
The bound is m > c n (log^3 n + k) where k depends on the convergence error
and n is the number of entries in the unknown signal. Even with modest c
and k values the number of equations m might grow very quickly compared to n.
In practice this might make the algorithm quite impractical.
We can see in sec 6 at lines 428-9 that a modest m = 6n is used.
However, do the constant vary much with real data?

I would have liked to see a discussion of the authors on this aspect.


Clarity
--
The paper is generally well-written and makes an effort in explaining
the general idea either before or after a result has been presented.
However, I have still found it quite heavy. Also, please provide
all the information needed to replicate your experiments.



Originality
--
The proposed analysis is novel in my opinion.
The algorithm perhaps not.


Significance
--
The fact that this simple modification of the alternating minimization
algorithm converges might be useful also to similar problems.
I suppose that other researchers might try this algorithm
on their problems.





Summary: The paper provides some new analysis to techniques similar to projections-based optimization.
The results seem more original than the algorithm itself. However, the analysis is extremely
lengthy (17 pages in the supplementary material).

Submitted by Assigned_Reviewer_5

In this paper, the authors present a new algorithm for phase retrieval from magnitude measurements. They provide analysis about the quality of their new initialization and the subsequent iterative steps. The paper is technically deep but quite clear. They compare their method against that of existing phase retrieval algorithms, although they restrict the comparisons to only Gaussian measurements. The numerical results compare their algorithm to that of PhaseCut and PhaseLift, two other algorithms. These competing algorithms work on Gaussian and non-Gaussian measurements. Do the authors' results extend to non-Gaussian measurements?

In the numerical results, one important plot that is missing is how much error reduction happens after the initialization phase? It would be interesting to see if the geometric error reduction on Theorem 4.2 occurs in practice. Can the authors comment on that or add an experiment to this effect?
Summary: A new algorithm is presented for phase retrieval, with good analysis and reasonable numerical experiments.

Submitted by Assigned_Reviewer_6

The paper studies some theoretical guaranties of the classical alternating minimization algorithms used for Phase Retrieval.

The paper is well written, and much more interesting than the title suggests (well, I prefer that than the opposite).

The framework considered here, is the recent framework proposed by Candes et al. where the measurement matrix is random. Such a framework allows theoretical studies, which are much more difficult in the general case. However, I have still some doubt of the real "practical" effect of such a framework.
Indeed, phase retrieval is an old physical problem which arises when the physical instruments lose the phase information. Random sensing matrices appears only in the compressed sensing framework. So, I wonder which are real applications of the phase retrieval problem with random matrices.
I think that a short discussion about that would be relevant in the paper.

However, the theoretical questions addressed here are still relevant and interesting. The structure of the paper is clear, and I have no particular remarks on this aspect.

My other minor concern is about the first sentence of the abstract: "Phase retrieval problems involve solving linear equations". Phase retrieval problem is a typical problem of solving *non*-linear equations, because of the modulus !





Summary: A pleasant paper about phase retrieval, dealing with random matrices instead of "physical" matrices.
Author Feedback

Author rebuttal: We would like to thank the reviewers for their detailed comments. We first respond to the reviewers' comments about the practicality of the present approach and its performance on non-Gaussian measurement schemes.

Performance on non-Gaussian measurement schemes: While phase retrieval is traditionally studied for Fourier measurements, so far no algorithm has provable recovery in this setting. To the best of our knowledge, existing methods like PhaseLift and PhaseCut, as well as our proposed method, have theoretical guarantees only for Gaussian measurements. However, in [4], the authors propose a physical measurement scheme using masks and show that PhaseLift works well (empirically) for random binary and Gaussian masks. Figure 2(a) shows that our algorithm also achieves exact recovery for Gaussian masks. We have observed that our algorithm works well with random binary masks as well - we can include a plot of the same in the final version.

Assigned_Reviewer_4:

Splitting of equations: We would like to stress that, in practice, we use the whole set of equations in each iteration (i.e., Algorithm 1). However in the analysis, we use subsets of equations (i.e., Algorithm 2) to remove dependence between two different iterates (i.e., x^t and x^{t+1}), which makes the analysis significantly easier. Obtaining theoretical guarantees for Algorithm 1 is a (challenging) open problem.

Value of the constants: We have not tried to optimize the constants in our analysis but have instead focused on the scaling behavior. Our experiments suggest that, in practice, the constants in our results seem to be quite small. For instance, from Figure 1, we see that Algorithm 1 requires around 5n measurements for successful recovery. We have also observed that Algorithm 2 requires about 5n log(1/epsilon) measurements for successful recovery (i.e., 5n measurements in each iteration).

Replication of experiments: We forgot to mention that we choose $x^*$ uniformly at random from the unit sphere. Apart from this, we believe we have given all information necessary to replicate our experiments. If the reviewer thinks we missed something, we will be happy to include it in the final version.

Lines 308-309: There is a typo in Lines 308-309 - the term in the middle should have dist(x,x*)^2 instead of dist(x,x*) in the numerator. To prove the inequality, let x+ = (alpha) x* + (beta) z where < z,x* > = 0. Then, dist(x+,x*)^2 = beta^2 / (alpha^2 + beta^2) \leq beta^2 / alpha^2, giving the first inequality. The second inequality follows if 25/ 81(1 - c dist(x,x*))^2 < 9 / 16. This is true if dist(x,x*) is smaller than some absolute constant.
Line 311 is a typo.
Lines 190-191: C should be Diag(Ph(A^T x)).
Line 199: Yes, the expression for S should have y_i^2 instead of y_i.

Assigned_Reviewer_5:

Q: Plots for reduction in error after each iteration.
A: We do see geometric decay of error in each iteration of alternating minimization. We will be happy to include a plot of the same in the final version/supplementary material.

Assigned_Reviewer_6:

Q: My other minor concern...
A: Duly noted.